# How reliable are self-reported estimates of birth registration completeness? Comparison with vital statistics systems

**Tim Adair**[1]*, **Alan D. Lopez**[2]

1 Melbourne School of Population and Global Health, The University of Melbourne, Carlton, Victoria, Australia, 2 Institute of Health Metrics and Evaluation, University of Washington, Seattle, WA, United States of America

* timothy.adair@unimelb.edu.au

## Abstract

### Background

The widely-used estimates of completeness of birth registration collected by Demographic and Health Surveys (DHS) and Multiple Indicator Cluster Surveys (MICS) and published by UNICEF primarily rely on registration status of children as reported by respondents. However, these self-reported estimates may be inaccurate when compared with completeness as assessed from nationally-reported official birth registration statistics, for several reasons, including over-reporting of registration due to concern about penalties for non-registration. This study assesses the concordance of self-reported birth registration and certification completeness with completeness calculated from civil registration and vital statistics (CRVS) systems data for 57 countries.

### Methods

Self-reported estimates of birth registration and certification completeness, at ages less than five years and 12–23 months, were compiled and calculated from the UNICEF birth registration database, DHS and MICS. CRVS birth registration completeness was calculated as birth registrations reported by a national authority divided by estimates of live births published in the United Nations (UN) World Population Prospects or the Global Burden of Disease (GBD) Study. Summary measures of concordance were used to compare completeness estimates.

### Findings

Birth registration completeness (based on ages less than five years) calculated from self-reported data is higher than that estimated from CRVS data in most of the 57 countries (31 countries according to UN estimated births, average six percentage points (p.p.) higher; 43 countries according to GBD, average eight p.p. higher). For countries with CRVS completeness less than 95%, self-reported completeness was higher in 26 of 28 countries, an average 13 p.p. and median 9–10 p.p. higher. Self-reported completeness is at least 30 p.p. higher than CRVS completeness in three countries. Self-reported birth certification completeness exhibits closer concordance with CRVS completeness. Similar results are found for self-reported completeness at 12–23 months.

**Data Availability Statement:** All relevant data are within the paper and its Supporting Information files.

**Funding:** This study was funded under an award from Bloomberg Philanthropies and the Australian

Department of Foreign Affairs and Trade to the University of Melbourne to support the Data for Health Initiative. The funders had no role in study design, data collection and analysis, decision to publish, or preparation of the manuscript.

**Competing interests:** The authors have declared that no competing interests exist.

## Conclusions

These findings suggest that self-reported completeness figures over-estimate completeness when compared with CRVS data, especially at lower levels of completeness, partly due to over-reporting of registration by respondents. Estimates published by UNICEF should be viewed cautiously, especially given their wide usage.

## Introduction

Complete birth registration and certification within a civil registration and vital statistics (CRVS) system provides major benefits for both individuals and societies. It ensures legal identity for individuals and provides them with citizenship and voting rights, access to social security benefits and health and education services, proof of age, and, above all, has been described as a fundamental human right [1–7]. Complete birth registration, where births are registered in a timely manner (i.e. within one year of the birth), should also be the primary source of fertility statistics to track trends in birth rates, provide denominators to calculate early age mortality rates, serve as the fundamental input into population projections, and inform government planning for health (e.g. childhood vaccinations), education and social services [1]. The importance of birth registration for development policies is demonstrated by its critical role in monitoring progress towards Sustainable Development Goal 16.9, which aims to provide legal identity for all, including birth registration, by 2030 [8, 9].

Birth registration, however, is incomplete in many low- and middle-income countries [10–12]. Regular measurement of birth registration completeness using reliable and consistent methods enables countries and development partners to monitor progress towards development goals, including achieving universal birth registration, and also to adjust fertility statistics produced by birth registration data, which have a number of policy uses across many sectors of government. Birth registration completeness can be calculated as the number of births registered in a timely manner reported by a national authority, divided by an estimate of the total number of births such as the estimates routinely published in the United Nations (UN) World Population Prospects or, more recently, by the Global Burden of Disease (GBD) Study [13, 14].

Estimates of birth registration completeness are produced by UNICEF and published annually in their *The State of the World's Children* reports [11]. The latest report, published in 2019, estimated that 27% of children aged less than five years have had not their birth registered. UNICEF relies on a range of data in compiling their estimates, which, in countries with incomplete birth registration, predominantly comes from data collected in major survey platforms such as the Demographic and Health Surveys (DHS), Multiple Indicator Cluster Surveys (MICS) or other national surveys. In these surveys, the respondent (a parent or caregiver) is asked whether or not a child aged less than five years at the time of the survey has a birth certificate or if their birth has been registered [11]. This definition of completeness—the percentage of children less than five years whose birth is registered—is consistent with that used by Sustainable Development Goal 16.9 [9].

However, for a number of reasons, the use of self-reported data and the definition of birth registration completeness used by the DHS and MICS and employed in *The State of the World's Children* reports (and also more broadly by UNICEF) may result in inaccurate measurement of the completeness of the timely registration of births. Firstly, self-reported registration data may be subject to over-reporting by the respondent, i.e. reporting that the birth was registered when it was not. Inaccurate reporting may occur because of genuine confusion about whether the birth was registered or concern about being penalised for not having

registered the birth. For example, in one survey in Rwanda the family could only provide a birth certificate in 10% of births reported to be 'registered', which raises concerns about the accuracy of birth registration information provided in these surveys [12]. The provision of evidence of a birth certificate during the interview is not a pre-requisite to measure the birth as being registered, likely because the certificate may not be readily accessible during the interview. The enumerator may also mistakenly regard an incorrect document as evidence of birth registration, as was found to have occurred in a Census in Mali where the incorrect document was a family card used for taxes [15]. Secondly, birth certification completeness (the percentage of children reported to have a birth certificate, whether or not seen by the interviewer) may provide a more reliable measurement of birth registration completeness because it is a reference point for the respondent knowing whether the birth was registered, and in many cases evidence of the certificate is provided to the interviewer. However, this may potentially underestimate completeness if not all registered births are issued a certificate. Additionally, self-reported birth certification is also subject to potentially inaccurate reporting by the respondent in cases where they do not provide evidence of the certificate.

Thirdly, even if respondents accurately report birth registrations, this does not mean that the data on birth registration were transferred and consolidated at the national level for reporting of birth statistics. Fourthly, some births may be registered more than one year after the birth, for example when the child is about to commence schooling. Delayed birth registration is therefore not timely either for the child (e.g. to access essential health services) or to provide reliable statistics. The proportion of registered births of children under five years that were registered before they turned one year of age would vary by country. Additionally, information on the registration of births of older children may be more subject to recall bias by respondents. And finally, self-reported registration data are only provided for children alive at the time of the survey. Children that have died are more likely to be from lower socio-economic groups and therefore less likely to have had their birth registered, meaning that completeness estimates based on live children are likely to be an overestimate. Mortality of children before their birth is registered would also be more likely where the registration of the birth is delayed. Also, babies who die in the neonatal period are commonly not registered and so this may also inflate birth registration completeness estimates [1].

Given these limitations of the self-reported birth registration completeness estimates, and the widespread reliance on these estimates for policy and planning, it is of interest to assess their reliability against nationally-reported official birth registration statistics derived from CRVS data. We make use of the recent compilation and publication of a global birth registration database and corresponding estimates of birth registration completeness to compare the self-reported birth registration and certification completeness estimates for 57 countries [10]. We contrast results for CRVS completeness calculated using UN and GBD birth estimates. Given the limitations with self-reported birth registration data detailed above, we might reasonably expect that the statistics that use these data will over-estimate birth registration completeness compared with CRVS data. Further, to assess the extent to which delayed birth registration contributes to differences in self-reported and CRVS completeness, we also re-calculate self-reported completeness for children ages 12 to 23 months.

## Methods

### Completeness–Self-reported data

Self-reported data on birth registration were primarily taken from the 2019 *State of the World's Children* report and available in the UNICEF birth registration database [11, 12, 16]. From this database we used Demographic and Health Survey (DHS), Multiple Indicator Cluster Survey

(MICS) and other survey or census data from 2006 onwards that published self-reported birth registration data; the UNICEF database also includes vital registration data but these were not included in our study because of our focus on self-reported data. We also compiled self-reported birth registration data from countries whose data were published subsequent to the release of the UNICEF database by using the *DHS STATcompiler* tool or individual survey publications (see S1 Text) [17]. In total there were 119 countries with self-reported birth registration data; 116 from the UNICEF database and three from surveys that were published subsequently.

Birth registration completeness based on self-reported data is measured as the percentage of children aged less than five years of age having a birth certificate, or whose birth was reported in the survey as having been registered with the civil authorities [12]. This is derived from:

1. a question that asks the respondent (a family member or carer) to report, for each child in the household, whether they have a birth certificate, and if the interviewer has seen the certificate; and

2. if the child is not reported to have a birth certificate, a follow-up question that asks the respondent whether the child's birth was registered with a civil authority.

A child is considered to have had their birth registered if the respondent reported that he or she has a birth certificate, regardless of whether or not the certificate was seen by the interviewer, or if the birth was registered.

We also measured self-reported birth certification completeness as the percentage of children under age five years with a birth certificate, whether seen or not by the interviewer, where this indicator was reported in *DHS STATcompiler*, in the individual survey publication or could be calculated from the DHS or MICS microdata (see S1 Text) [17–19]. Most surveys do not publish whether the birth certificate was seen or not by the interviewer, possibly because the certificate may not be readily accessible to respondents during the interview.

One limitation of the measurement of self-reported birth registration completeness for children aged less than five years is that births with untimely registration are included. To assess whether this may contribute to differences with CRVS completeness estimates (calculated from UN birth estimates), we also calculated self-reported birth registration and certification completeness for children aged 12 to 23 months; use of this age group allows for registration within one year of their birth but excludes births registered from two years of age onwards where registration would be, by any definition, untimely. We analysed DHS or MICS microdata to calculate self-reported completeness for ages 12–23 months (see S1 Text) [18, 19].

## Completeness–CRVS data

CRVS birth registration data includes available birth registration data reported by a national authority. Such data were available for 62 of the 119 countries that have self-reported birth registration completeness estimates; there were no CRVS data available for the other 57 countries. These CRVS data are primarily from a global database of birth registration published as part of a global assessment of the utility of birth registration data [10]. These comprise data reported to the United Nations Statistical Division (UNSD) by countries in standardized tables in the Demographic Yearbook questionnaire, as well as data published or made available by countries that are not in the UNSD database [20, 21]. This database was updated with additional data that were not available at the time of the database's publication. A limitation of these data is that there is not always information about whether births are reported by year of occurrence or year of registration, or whether births which were registered late (e.g. 1 year or more after

occurrence) are included. Where possible, we used data on births that occurred in the calendar year and were registered within one year of the birth. Birth registration data for eight of these countries are unpublished and were made available to the authors through established collaborations; for these countries, absolute differences with self-reported completeness are presented.

Birth registration completeness according to the CRVS data was calculated as the number of registered births divided by the number of estimated live births reported in the UN World Population Prospects and also in the GBD Study [13, 14]. In countries with incomplete birth registration, both the UN World Population Prospects and GBD estimate live births predominantly from census and survey data using demographic and statistical models. The GBD and UN do use birth registration as a source of fertility estimates where such data are complete, which may create dependence between the numerator and denominator; we therefore filter our analyses to countries with completeness less than 95% (see below). The estimates of completeness of birth registration according to CRVS data may be biased if the number of live births estimated by the UN World Population Prospects or GBD is inaccurate. Where the number of registered births exceeded the number of estimated births, we assumed completeness of 100%. We used CRVS birth registration completeness for the year closest to the mid-point of the quinquennial period preceding the date of the source of self-reported birth registration data (because completeness was measured for children aged less than five years); we excluded CRVS data which were more than 10 years older than this mid-point.

## Comparison of self-reported and CRVS completeness

Our comparison of self-reported and CRVS estimates of birth registration completeness was conducted for all countries where both estimates of birth registration completeness data were available. Of the 62 countries with self-reported birth registration data, 57 countries also had CRVS birth registration data within the 10-year time frame defined above and so were included in our study (5 countries had CRVS data outside of the time frame) (see Table 3). Of the 57 countries included in our study, 44 countries had data on birth certification completeness.

About half (29) of the 57 study countries had birth registration completeness according to the CRVS data of at least 95%. In these countries it is very likely that differences with self-reported registration completeness will be low and, because they have virtually universal birth registration, self-reported data are less likely to be used than for countries where the CRVS data on births are less complete. We therefore separately analysed the 28 countries (22 with birth certification completeness) where birth registration completeness, according to the CRVS data, was less than 95%. These calculations were conducted with CRVS completeness calculated using both UN and GBD birth estimates. We were able to calculate self-reported registration completeness for ages 12–23 months for 39 of the 57 countries (data were not available for the other 18 countries); for 37 of these countries we could also calculate completeness based on certification.

To compare CRVS and self-reported completeness we calculated the mean and median absolute difference and root squared difference in percentage points. In these summary results we included the results for the eight countries with unpublished data. In nine countries, the UNICEF birth registration database states that birth registration completeness differs from the standard definition or refers to only part of a country; for these countries we separately measured concordance with CRVS birth registration completeness [16].

## Results

Fig 1 compares the completeness of birth registration according to the CRVS data (using both UN and GBD birth estimates) with completeness of birth registration and certification from

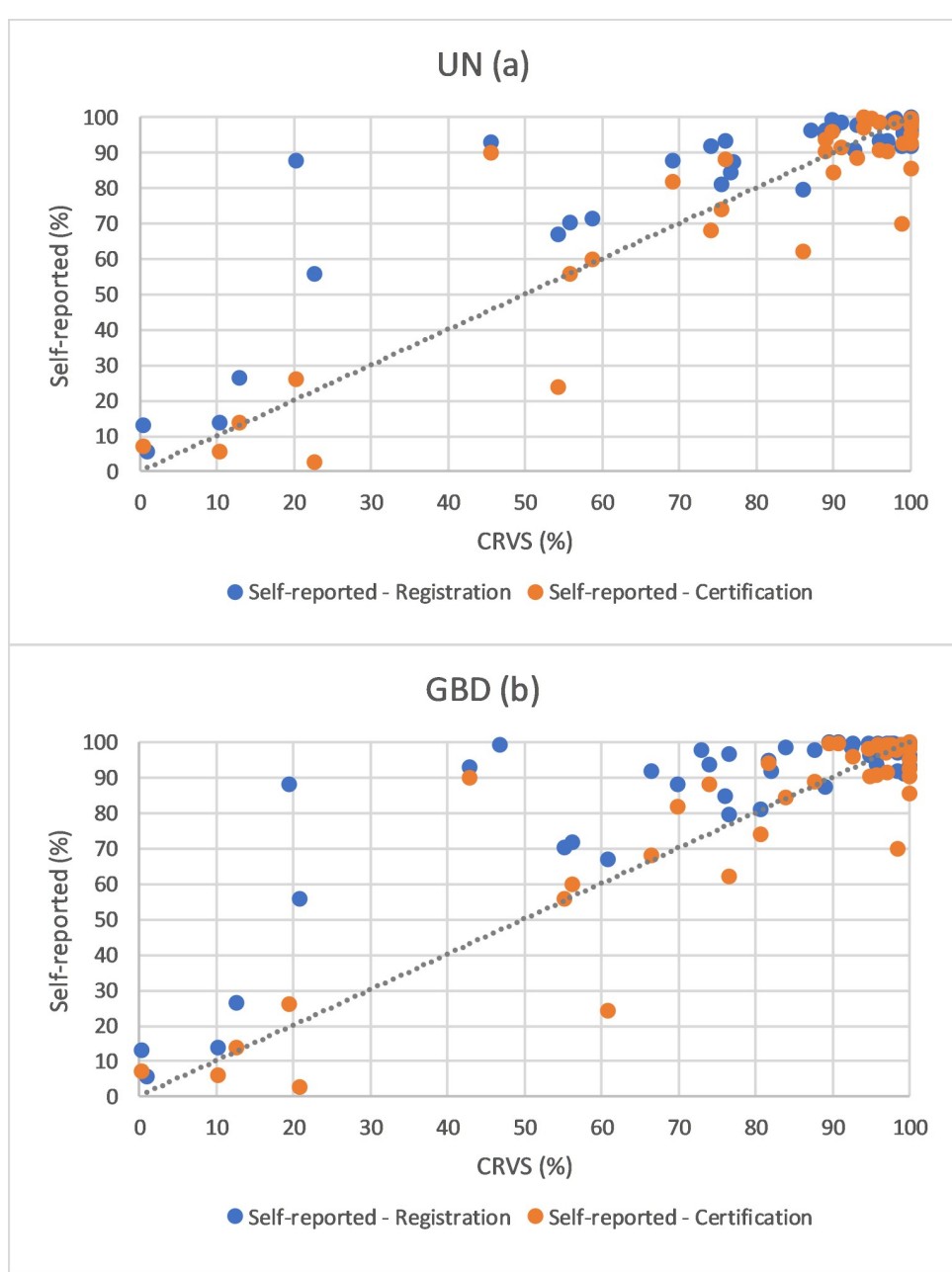

**Fig 1.** Comparison of completeness of birth registration of CRVS data (UN (a) and GBD (b) estimated births) and completeness of birth registration and certification from self-reported data (%).

the self-reported data. Self-reported completeness is consistently higher than that reported by the CRVS data, particularly among countries at lower levels of registration. The four countries with substantially higher self-reported than CRVS completeness are Rwanda, Lebanon, Solomon Islands and Paraguay (further information provided below). Certification completeness according to the self-reported data are, overall, closer to the CRVS registration completeness figures, but with some countries with much lower self-reported completeness (Rwanda, Kenya, India and Saint Lucia).

**Table 1. Summary comparison metrics for completeness from CRVS data (calculated using UN birth estimates) and completeness from self-reported data, less than five years.**

| Scope (countries) | CRVS–registration completeness (%) | | Higher completeness (number of countries)* | | Absolute difference (Self-reported minus CRVS) (percentage points) | | Root squared difference (percentage points) | |
|---|---|---|---|---|---|---|---|---|
| | Mean | Median | CRVS | Survey | Mean | Median | Mean | Median |
| **Self-reported: registration completeness^** | | | | | | | | |
| All countries (57) | 82.3 | 95.0 | 25 | 31 | +5.9 | +1.8 | 7.7** | 4.8 |
| All countries with CRVS completeness less than 95% (28) | 64.8 | 76.4 | 2 | 26 | +13.0 | +8.9 | 13.6 | 8.9 |
| **Self-reported: certification completeness^^** | | | | | | | | |
| All countries (44) | 81.2 | 94.5 | 28 | 16 | -1.5 | -0.9 | 6.8 | 4.3 |
| All countries with CRVS completeness less than 95%^^^ (22) | 63.2 | 74.8 | 9 | 13 | +0.5 | +1.2 | 9.2 | 5.5 |

*In some countries there was no difference between self-reported and CRVS completeness estimates.

**Root mean squared difference is 4.4 percentage points for the 9 countries for which UNICEF state that the data differ from the standard definition or refer to only part of a country.

^ Self-reported registration completeness: The percentage of children aged less than five years having a birth certificate (whether or not seen by the interviewer), or whose birth was reported in the survey as having been registered with the civil authorities.

^^ Self-reported certification completeness: The percentage of children aged less than five years having a birth certificate (whether or not seen by the interviewer).

^^^ As measured using UN estimated births.

There were 57 countries with both CRVS and self-reported registration completeness estimates; using UN estimated births, the self-reported completeness was higher in 31 countries and the CRVS completeness higher in 25 countries, with the respective figures using GBD estimated births being 43 and 13 (Tables 1 and 2). The self-reported registration completeness was higher on average (UN: 5.9 percentage points higher, GBD: 8.3 higher) and the root mean squared difference was 7.7 percentage points (median 4.8) according to UN estimated births and 9.7 (median 4.4) according to GBD estimated births. Importantly, when only the 28 countries with CRVS completeness less than 95% were included, the differences are much greater.

**Table 2. Summary comparison metrics for completeness from CRVS data (calculated using GBD birth estimates) and completeness from self-reported data, less than five years.**

| Scope (countries) | CRVS–registration completeness (%) | | Higher completeness (number of countries)* | | Absolute difference (Self-reported minus CRVS) (percentage points) | | Root squared difference (percentage points) | |
|---|---|---|---|---|---|---|---|---|
| | Mean | Median | CRVS | Survey | Mean | Median | Mean | Median |
| **Self-reported: registration completeness^** | | | | | | | | |
| All countries (57) | 79.9 | 94.6 | 13 | 43 | +8.3 | +3.0 | 9.7* | 4.4 |
| All countries with CRVS completeness less than 95%* (28) | 64.5 | 76.2 | 2 | 26 | +13.4 | +10.3 | 14.1 | 10.3 |
| **Self-reported: certification completeness^^** | | | | | | | | |
| All countries (44) | 80.1 | 93.6 | 19 | 25 | -0.5 | +0.6 | 7.4 | 4.2 |
| All countries with CRVS completeness less than 95%^^^ (22) | 62.3 | 71.9 | 7 | 15 | +1.5 | +1.2 | 9.7 | 6.0 |

* Root mean squared difference is 5.4 percentage points for the 9 countries for which UNICEF state that the data differ from the standard definition or refer to only part of a country.

^ Self-reported registration completeness: The percentage of children aged less than five years having a birth certificate (whether or not seen by the interviewer), or whose birth was reported in the survey as having been registered with the civil authorities.

^^ Self-reported certification completeness: The percentage of children aged less than five years having a birth certificate (whether or not seen by the interviewer).

^^^ As measured using UN estimated births.

Self-reported completeness was higher in 26 of the 28 countries, on average by 13 percentage points, with a root mean squared difference of 14 percentage points (for both UN and GBD; UN median 8.9, GBD median 10.3). For the nine countries for which UNICEF state that the data differ from the standard definition or refer to only part of a country, the root mean squared difference was only 4.4 percentage points for UN estimated births and 5.4 for GBD estimated births.

Self-reported certification completeness, compared with CRVS registration completeness according to UN estimated births, was higher in 16 of 44 countries, and according to GBD estimated births was higher in 25 countries. Self-reported certification completeness was on average lower than CRVS registration completeness (-1.5 percentage points, GBD: -0.5) with a root mean squared difference of 6.8 percentage points (median 4.3) according to UN estimated births and 7.4 (median 4.2) according to GBD estimated births. For the 22 remaining countries with CRVS registration completeness less than 95%, self-reported certification completeness was higher in most countries (UN: 13, GBD: 15) with small mean absolute differences (UN: +0.5 percentage points, GBD: 1.5) and a root mean squared difference of almost 10 percentage points (UN: 9.2, median 5.5; GBD: 9.7, median 6.0). For all 57 countries, average CRVS birth registration completeness using the GBD birth estimates (79.9%) was lower than based on UN birth estimates (82.3%).

Tables 3 and 4 shows the results for each country. The largest differences are found in three countries where self-reported completeness exceeded CRVS completeness by 30 percentage points; in Solomon Islands (UN 68 percentage points, GBD 69 percentage points), Paraguay (UN 48 percentage points, GBD 50 percentage points), and Rwanda (UN 33 percentage points, GBD 35 percentage points). In some cases, CRVS completeness is higher than what was self-reported, with the most extreme example being Bolivia with CRVS completeness using UN estimated births (self-reported 92%, CRVS (UN) 100%). A notable wide discrepancy in completeness between GBD and UN birth estimates is found in Lebanon.

For the 39 countries where self-reported completeness could be calculated for children aged 12–23 months, self-reported completeness differed only slightly from completeness calculated for children less than five years (Table 5). In fact, completeness was slightly higher among children 12–23 months (88.1% versus 87.5%). As a result, differences with CRVS completeness (using UN estimated births) were similar to those calculated in Table 1, with a mean absolute difference for 12–23 years of +6.8 percentage points (+5.9 for less than five years) and root mean squared difference of 7.4 percentage points (7.7 for less than five years). Such similar results were also found for when excluding countries with CRVS completeness of at least 95% and when comparing self-reported certification completeness. Individual country results are presented in S1 and S2 Tables.

## Discussion

Information routinely collected by DHS and MICS and compiled by UNICEF on the basis of self-reports about whether or not the births of surviving children have been certified or registered is widely used to inform health and social planning. Yet, our analysis suggests that self-reported birth registration data as reported by UNICEF over-estimates completeness compared with available national registration data. Excluding countries where birth registration completeness based on CRVS data is at least 95% (i.e. may be considered as complete), birth registration completeness calculated from self-reported data is higher than that suggested by CRVS data, calculated using either UN or GBD estimated births, in 26 out of 28 countries and being an average 13 percentage points higher and median of 9–10 percentage points higher. This difference is less extreme for countries where at least 95% of births are registered, since

**Table 3. CRVS completeness (calculated both using UN and GBD birth estimates) and self-reported completeness (%), less than five years, by country.**

| Countries and areas | CRVS registration completeness | | Self-reported completeness (%) | | CRVS data year | Self-reported data source, year |
|---|---|---|---|---|---|---|
| | UN | GBD | Certification | Registration | | |
| Albania | 90 | 84 | 85 | 98 | 2013 | DHS 2017–18 |
| Argentina* | 100 | 99 | 99 | 100 | 2009 | MICS 2011–2012 |
| Armenia | 100 | 100 | 99 | 99 | 2014 | DHS 2015–2016 |
| Azerbaijan | 76 | 74 | 88 | 94 | 2003 | DHS 2006 |
| Barbados | 100 | 98 | 98 | 99 | 2007 | MICS 2012 |
| Bhutan | 95 | 91 | 100 | 100 | 2005 | MICS 2010 |
| Bolivia (Plurinational State of)* | 100 | 82 | – | 92 | 2013 | EDSA 2016 |
| Bosnia and Herzegovina | 100 | 96 | – | 100 | 2004 | MICS 2006 |
| Botswana* | 77 | 89 | – | 88 | 2017 | Demographic Survey 2017 |
| Cabo Verde | 93 | 99 | – | 91 | 2017 | Censo 2010 |
| Colombia | 90 | 77 | – | 97 | 2013 | DHS 2015 |
| Côte d'Ivoire | 59 | 56 | – | 72 | 2013 | MICS 2016 |
| Cuba | 100 | 100 | 100 | 100 | 2012 | MICS 2014 |
| Dominican Republic | 69 | 70 | – | 88 | 2012 | MICS 2014 |
| Egypt | 100 | 100 | 99 | 99 | 2012 | DHS 2014 |
| El Salvador | 100 | 100 | 86 | 99 | 2012 | ENS/MICS 2014 |
| Georgia | 100 | 98 | – | 100 | 2013 | WMS 2015 |
| Guatemala* | 87 | 95 | – | 96 | 2012 | ENSMI 2014–2015 |
| Honduras | 96 | 96 | 91 | 94 | 2010 | DHS 2011–2012 |
| India | 86 | 76 | 62 | 80 | 2013 | NFHS 2015–16 |
| Iran (Islamic Republic of)* | 98 | 92 | – | 99 | 2011 | MIDHS 2010 |
| Jordan | 93 | 88 | 89 | 98 | 2015 | DHS 2017–2018 |
| Kazakhstan | 100 | 98 | 100 | 100 | 2013 | MICS 2015 |
| Kenya | 54 | 61 | 24 | 67 | 2012 | DHS 2014 |
| Kyrgyzstan | 94 | 97 | 97 | 99 | 2015 | MICS 2018 |
| Lebanon | 98 | 47 | – | 100 | 2008 | MICS 2009 |
| Maldives | 91 | 97 | 92 | 99 | 2014 | DHS 2016–2017 |
| Mexico | 89 | 82 | 94 | 95 | 2012 | MICS 2015 |
| Mongolia | 100 | 97 | 99 | 100 | 2015 | MICS 2018 |
| Montenegro | 100 | 100 | 93 | 96 | 2009 | MICS 2018 |
| Nicaragua | 77 | 76 | – | 85 | 2010 | ENDESA 2011/2012 |
| North Macedonia | 100 | 95 | 98 | 100 | 2009 | MICS 2011 |
| Panama | 99 | 100 | 93 | 96 | 2011 | MICS 2013 KFR |
| Paraguay | 45 | 43 | 90 | 93 | 2013 | MICS 2016 |
| Peru* | 100 | 73 | – | 98 | 2014 | ENDES 2016 prelim |
| Philippines | 74 | 66 | 68 | 92 | 2015 | DHS 2017 |
| Republic of Moldova | 90 | 93 | 96 | 100 | 2010 | MICS 2012 |
| Saint Lucia | 99 | 98 | 70 | 92 | 2013 | MICS 2012 |
| Serbia | 100 | 99 | – | 99 | 2012 | MICS 2014 |
| Sri Lanka | 100 | 98 | – | 97 | 2006 | DHS 2006–2007 |
| Suriname | 100 | 100 | 95 | 98 | 2014 | MICS 2018 |
| Thailand* | 100 | 99 | 100 | 100 | 2011 | MICS 2015–2016 |
| Tonga | 97 | 100 | 91 | 93 | 2003 | DHS 2012 |
| Trinidad and Tobago | 89 | 95 | 91 | 97 | 2009 | MICS 2011 |
| Tunisia | 100 | 100 | 98 | 100 | 2011 | MICS 2018 |

*(Continued)*

**Table 3.** (Continued)

| Countries and areas | CRVS registration completeness | | Self-reported completeness (%) | | CRVS data year | Self-reported data source, year |
|---|---|---|---|---|---|---|
| | UN | GBD | Certification | Registration | | |
| Turkey* | 96 | 97 | 99 | 99 | 2011 | DHS 2013 |
| Ukraine | 100 | 96 | 99 | 100 | 2010 | MICS 2012 |
| Uruguay | 98 | 98 | 99 | 100 | 2012 | MICS 2013 |
| Uzbekistan | 94 | 90 | – | 100 | 2005 | MICS 2006 |

* Data for which UNICEF state "Data differ from the standard definition or refer to only part of a country" [16].

the methodological limitations of self-reported data on live children are likely to be less important in these populations. Of concern, self-reported completeness was over 30 percentage points higher than CRVS completeness in three countries (Paraguay, Rwanda, Solomon Islands). Self-reported completeness re-calculated for children aged 12–23 months is in fact marginally higher on average than when measured for children less than five years, despite it excluding births registered at least two years after occurrence. There is a smaller difference between self-reported birth certification completeness and CRVS birth registration completeness, with the mean absolute difference being less than one percentage point and root mean squared difference nine percentage points.

These findings suggest that estimates of birth registration completeness based on self-reported data collected by DHS and MICS, and routinely published by UNICEF, should be viewed cautiously. In particular, although the *State of the World's Children* reports, largely based on self-reported data. that 73% of children aged less than five years have had their birth registered, the actual level of birth registration completeness, where births are registered within one year of the birth, is likely to be significantly lower [11]. It is difficult to disentangle the specific reasons for the differences. One likely contributor is over-reporting of birth registration by respondents where they knew it should have been registered, even it was not, because of

**Table 4. Absolute difference (self-reported completeness minus CRVS completeness) calculated using UN and GBD birth estimates (percentage points), less than five years, by countries with unpublished data.**

| Countries | Absolute difference (Self-reported minus CRVS) (percentage points) | | | |
|---|---|---|---|---|
| | Self-reported certification | | Self-reported registration | |
| | UN | GBD | UN | GBD |
| Ghana | +0 | +1 | +15 | +16 |
| Malawi* | – | – | +5 | +5 |
| Myanmar | -2 | -7 | +5 | 0 |
| Papua New Guinea | +7 | +13 | +7 | +13 |
| Rwanda | -20 | -18 | +33 | +35 |
| Solomon Islands | +6 | +7 | +68 | +69 |
| United Republic of Tanzania | +1 | +1 | +13 | +13 |
| Zambia | -4 | +4 | -4 | +4 |

Authors' calculations. Country-years are Ghana: CRVS 2014, DHS 2014; Malawi: CRVS 2014, MICS 2013–14; Myanmar: CRVS 2013, DHS 2015–16; Papua New Guinea: CRVS 2015, DHS 2016–18; Rwanda: CRVS 2015, DHS 2014–15; Solomon Islands: CRVS 2014, DHS 2015; United Republic of Tanzania: CRVS 2013, DHS 2015–16; Zambia: CRVS 2014, DHS 2018.

* For self-reported data, UNICEF state "Data differ from the standard definition or refer to only part of a country" [16].

**Table 5. Summary comparison metrics for completeness from CRVS data (calculated using UN birth estimates) and completeness from self-reported data, 12–23 months.**

| Scope (countries) | Self-reported completeness (%)* | | Higher completeness (number of countries)** | | Absolute difference (Self-reported minus CRVS) (percentage points) | | Root squared difference (percentage points) | |
|---|---|---|---|---|---|---|---|---|
| | Mean | Median | CRVS | Survey | Mean | Median | Mean | Median |
| **Self-reported: registration completeness** | | | | | | | | |
| All countries (39) | 88.1 (87.5) | 98.9 (98.3) | 8 | 23 | +6.8 | +4.9 | 7.4 | 4.9 |
| All countries with CRVS completeness less than 95% (20) | 77.7 (77.1) | 91.3 (92.4) | 0 | 20 | +13.4 | +10.2 | 13.4 | 10.2 |
| **Self-reported: certification completeness** | | | | | | | | |
| All countries (37) | 82.6 (81.9) | 94.4 (92.6) | 20 | 14 | -0.7 | -0.2 | 6.4 | 3.5 |
| All countries with CRVS completeness less than 95% (18) | 68.1 (67.3) | 84.3 (83.3) | 6 | 12 | +1.7 | +2.6 | 9.4 | 5.1 |

* Figures in brackets are for children aged less than five years.

**In some countries there was no difference between self-reported and CRVS completeness estimates.

worry about being penalised for non-registration. The extent of such misreporting however cannot be directly measured without a further study. The stronger concordance of birth certification completeness with CRVS completeness suggests that respondents' reporting of whether a birth was certified (irrespective of whether they can produce the certificate), rather than whether it was registered, may be the more reliable measure of true birth registration. This may be because presentation of the certificate by the respondent is evidence of registration, or due to issuance of a birth certificate (even if unable to be shown to the interviewer) being a reference point for the respondent knowing that the birth was registered. As shown in this study, in some countries there is much lower self-reported birth certification than registration completeness, which may be due to over-reporting of registration or an actual low proportion of registered births that are certified.

Completeness based on birth certification, rather than just registration, may also indicate that the registered birth data has progressed further through the CRVS system and so is more likely to have been transferred, compiled and published at the national level [22]. Unfortunately, this may be a reason for differences in self-reported and CRVS birth registration completeness, especially considering that less than half the countries (57 of 119) with self-reported registration data have nationally reported birth registration statistics within 10 years of the survey, which suggests a general lack of understanding of the policy utility of reliable fertility statistics. It appears that untimely registration is not a significant cause of discrepancies between CRVS and self-reported completeness, because self-reported completeness measured for children 12–23 months is marginally higher than less than five years. However, other possible reasons affecting differences between completeness at 12–23 months and less than five years are that inclusion of very young children (e.g. less than six months) in the calculation lowers completeness because there has been less time for their birth to be registered, that completeness has been increasing in the years preceding the survey (i.e. younger children being more likely to be registered than older children), or that recall bias affects the accuracy of data for older children. Finally, while it is unlikely that the self-reported data being reliant upon registration of children still alive would affect results significantly, because in most countries less than 5% of children die before the age of five years, we would still expect some upward bias due to the likely correlation between birth registration and child survival prospects [23].

A limitation of the study is that the findings were based on a limited number of countries for which there was birth registration data available to compare against self-reported data. It is possible that our findings might be biased by this sample of countries, and hence not

generalizable to all low- and middle-income countries. This can only be assessed once more birth registration data become available. Additionally, as mentioned, it is not possible to precisely measure the extent to which self-reported registration completeness without evidence of a birth certificate suffers from respondent bias without conducting a closer investigation. Another limitation is that for the published CRVS birth registration data there is a lack of consistency or lack of information on whether births are reported by year of occurrence, rather than year of registration, or on what definition of late registration was used; such information is necessary to understand the extent to which CRVS completeness estimates are biased. Also, the accuracy of birth registration completeness estimates from CRVS data is dependent on the accuracy of the UN World Population Prospects and GBD birth estimates. There is some uncertainty regarding the accuracy of the completeness estimates calculated using the UN and GBD estimated births because the two sets of estimates are derived from different analytical approaches that result in higher completeness where GBD estimated births are used when compared with UN estimated births (because GBD commonly estimates lower births) [13, 14]. Additionally, their estimates of total births are derived from age-specific fertility rates applied to age-specific population data of women of reproductive age that is also subject to uncertainty. As a result, there can be significant differences in estimated completeness between the two sources, as in Lebanon, however in most countries these are small. These limitations may also contribute to differences between self-reported and CRVS completeness. Finally, while we used the most recent self-reported completeness data, in many countries these were conducted at least five years ago and so birth registration may have improved in the ensuing years.

## Conclusion

The self-reported birth registration completeness data collected by DHS and MICS and used by UNICEF have met a number of policy needs in the over 100 countries where they have been collected, in particular to track progress towards Sustainable Development Goal 16.9 and to demonstrate socio-economic differences in registration completeness using a wealth index or other variables not readily available in published CRVS data [24–26]. However there are many advantages of using CRVS data; these data can measure birth registration completeness at the small area level more accurately than sample surveys and can be used to track progress and better target interventions to increase completeness in an era where efforts, such as the Bloomberg Philanthropies Data for Health Initiative, are being made to improve registration of vital events. Furthermore, the compilation and publication of CRVS data as timely statistics at the national level using clear and standardised definitions, and with data disaggregated by other important components of birth statistics such as maternal age, child sex, birth order, and birth weight, can significantly enhance their policy utility and promote further investments towards universal birth registration [10].

Our results suggest that self-reported birth registration completeness estimates published by UNICEF, in cases where completeness is less than 95%, are likely to be at least 10 percentage points higher than what timely birth registration completeness suggests. This has very significant implications for monitoring progress towards development goals and targets, several of which require reliable estimates of annual births.

## Supporting information

**S1 Table. CRVS completeness (calculated using UN birth estimates) and self-reported completeness (%), 12–23 months, by country.**
(DOCX)

**S2 Table. Absolute difference (self-reported completeness minus CRVS completeness) calculated using UN birth estimates (percentage points), 12–23 months, by country with unpublished data.**
(DOCX)

**S1 Text. Country publications.**
(DOCX)

## Acknowledgments

The authors wish to acknowledge Surender Pandey for his assistance in compiling the data used in this study.

## Author Contributions

**Writing – original draft:** Tim Adair.

**Writing – review & editing:** Tim Adair, Alan D. Lopez.

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
