## [Decision Letter · Decision Letter 0]

19 Mar 2021

PONE-D-21-03554

How reliable are self-reported estimates of birth registration completeness? Comparison with vital statistics systems

PLOS ONE

Dear Dr. Lopez,

Thank you for submitting your manuscript to PLOS ONE. After careful consideration, we feel that it has merit but does not fully meet PLOS ONE’s publication criteria as it currently stands. Therefore, we invite you to submit a revised version of the manuscript that addresses the points raised during the review process.

Both the reviewers and I believe that the paper provides an important contribution and addresses an relevant topic in Demography.  The manuscript is well-written and clear. There are few suggests and comments that I would like to be addressed in the revised version, please see detailed comments below. There a few clarification issues and also some points that demand a little bit more discussion. 

We look forward to receiving your revised manuscript.

Kind regards,

Bernardo Lanza Queiroz, Ph.D

Academic Editor

PLOS ONE

Journal Requirements:

2. In the Methods section please provide further clarification how results from right countries with unpublished data was collected [line 224].

Reviewers' comments:

Reviewer's Responses to Questions

**Comments to the Author**

1. Is the manuscript technically sound, and do the data support the conclusions?

Reviewer #1: Yes

Reviewer #2: Yes

Reviewer #3: Partly

2. Has the statistical analysis been performed appropriately and rigorously? 

Reviewer #1: Yes

Reviewer #2: Yes

Reviewer #3: Yes

3. Have the authors made all data underlying the findings in their manuscript fully available?

Reviewer #1: Yes

Reviewer #2: Yes

Reviewer #3: Yes

4. Is the manuscript presented in an intelligible fashion and written in standard English?

Reviewer #1: Yes

Reviewer #2: Yes

Reviewer #3: Yes

5. Review Comments to the Author

Reviewer #1: This paper address a very important topic about the difference between self-reported birth registration and registers. I believe this paper makes an important contribution to the literature on this topic.

Reviewer #2: A good, interesting, and timely paper that does what it says on the tin.

A couple of pointers/concerns that the authors may wish to take into account in any revision.

1) I understand the need to take on UNICEF for its role in presenting such aggregated data; however, the real target of the paper is on the data that UNICEF use in aggregating the data (MICS / DHS). I would rather the focus were on those; with UNICEF being a prime example of how those data are used and prone to misinterpretation.

2) In comparing completeness with UNWPP and GBD, not enough is made of how dependent the assessment of completeness is on the relaibiltiy of those series. Yes, Lebanon is picked out as the extreme; but it points to a more fundamental issue, about those underlying data. In many instances the resulting estimates of completeness differ by more than ten percentage points. Should the authors not be drawing some kind of conclusion about the relative utility of the two series, where the GBD seems to produce somewhat lower estimates of births (and hence higher completeness) than does the WPP? (Table 1 could be better formatted to make it clearer which series is which (the heads of the sub-sections get lost in the welter of data).

3) Around lines 113-118. Another important consideration is that in much of the global South, registration may also occur when (and be delayed until) a child needs to go to school, which may require proof of birth. However, such a delay obviously attenuates the utility of the registration data, since any child dying in the interim is not covered.

Minor: Line 501. Months. Not years.

Reviewer #3: The objective of this paper is to assess the reliability of self-reported estimates of birth registration completeness obtained from surveys. It compares self-reported estimates with estimates computed using birth registrations reported by a national authority and estimates of the number of live births (UN and GBD).

This paper is interesting and relevant. It is well written and clearly preented. Overall, I found the results plausible, but I am not (yet) entirely convinced that self-reported estimates overestimate birth registration completeness.

I would encourage you to better discuss the impact of the denominator on estimates of completeness calculated from CRVS systems. In some countries, the numbers of live births are estimated using the number of births reported in CRVS. So, the denominator and the numerator are not independent. It will probably be found in places with high level of completeness, so it should not alter your results, but I think it is worth mentioning. In other countries, numbers of live births are based mainly on fertility estimates and estimates of population size by age groups, which may not be very reliable. In case numbers of births are overestimated, this could lead to lower estimates of completeness. Again, it may not substantially influence your results, but you could discuss this possibility.

I also think you should discuss the differences you find in CRVS registration completeness between UN and GBD (Table 2). In some countries, differences are huge (e.g. Lebanon), and often are non-negligible (e.g. Bolivia, India, Colombia, Mexico, Philippines). This indicates that the estimates of the number of births is far from perfect.

You could also provide analyses without the outliers. With GBD estimates, Paraguay and Lebanon are clear outliers, and Paraguay is also an outlier with UN data. Actually, if you remove these two countries, differences remain, but are smaller. Some other countries with big differences are also very small, and it may be worth mentioning it.

I did not understand why countries in Table 3 are not presented in the same way as in Table 2.

I also think you could report results at several points in time for the countries. If we find strong variations over time in either source, this would suggest there are some issues with the date. Moreover, since monitoring progress in birth registration is mentioned as important topic in the introduction, your results would be all the more relevant.

More literature would be useful to understand in more detail why survey data may lead to overestimating completeness. Research by Hertrich and Rollet in Mali (in French) worked on self-reported estimates in census data, and found that these were overestimated because of a wrong understanding of the rules by a few interviewers. This may be relevant to your paper.

Other comments

Could you mention the countries where estimated births are greater than registerd births? (line 197)

6. PLOS authors have the option to publish the peer review history of their article (what does this mean?). If published, this will include your full peer review and any attached files.

Reviewer #1: No

Reviewer #2: No

Reviewer #3: No

---

## [Author Response · Author response to Decision Letter 0]

27 Apr 2021

PONE-D-21-03554

How reliable are self-reported estimates of birth registration completeness? Comparison with vital statistics systems

Responses to reviewers’ comments

We thank the editor for the opportunity to revise this manuscript. Please see our responses below.

We have revised the title page and file names to required format.

2. In the Methods section please provide further clarification how results from right countries with unpublished data was collected [line 224].

We have now added that the data for the eight countries with unpublished data “were made available to the authors through established collaborations”. We felt it most appropriate to add this statement in Lines 193-94. 

Reviewer #1: This paper address a very important topic about the difference between self-reported birth registration and registers. I believe this paper makes an important contribution to the literature on this topic.

We thank the reviewer for their positive review of the manuscript.

Reviewer #2: A good, interesting, and timely paper that does what it says on the tin.

A couple of pointers/concerns that the authors may wish to take into account in any revision.

1) I understand the need to take on UNICEF for its role in presenting such aggregated data; however, the real target of the paper is on the data that UNICEF use in aggregating the data (MICS / DHS). I would rather the focus were on those; with UNICEF being a prime example of how those data are used and prone to misinterpretation.

The reviewer makes a good point about making the distinction between the data collectors and the disseminators of the data (UNICEF). We now refer to the data as being self-reported estimates of completeness of birth registration collected by DHS and MICS and published by UNICEF. This change has been made in several places: Line 22-23, 90, 127, 133, 153, 214, 346, 366, 430, 445-46

2) In comparing completeness with UNWPP and GBD, not enough is made of how dependent the assessment of completeness is on the relaibiltiy of those series. Yes, Lebanon is picked out as the extreme; but it points to a more fundamental issue, about those underlying data. In many instances the resulting estimates of completeness differ by more than ten percentage points. Should the authors not be drawing some kind of conclusion about the relative utility of the two series, where the GBD seems to produce somewhat lower estimates of births (and hence higher completeness) than does the WPP? (Table 1 could be better formatted to make it clearer which series is which (the heads of the sub-sections get lost in the welter of data).

The reviewer makes a good point about the lower estimates of births (and therefore birth registration completeness) made by the GBD compared with the UN WPP. We have chosen to present completeness estimated using both UN WPP and GBD birth estimates because they are both prominent sources of country birth estimates. We do not however make conclusions about the relative utility of the two estimates because, firstly, it is not the primary aim of the manuscript (such an analysis is beyond the scope of our stated aims and would require a separate manuscript) but also that the two estimation processes utilize different approaches for estimating fertility from vital registration data (often incomplete), complete birth histories and summary birth histories. In the limitations (lines 416-423) we have pointed out that “There is some uncertainty regarding the accuracy of the completeness estimates calculated using the GBD and UN estimated births because the two sets of estimates are derived from different analytical approaches that result in higher completeness where GBD estimated births are used when compared with UN estimated births (because GBD commonly estimates lower births). Additionally, their estimates of total births are derived from age-specific fertility rates applied to age-specific population data of women of reproductive age that is also subject to uncertainty. As a result, there can be significant differences in estimated completeness between the two sources, as in Lebanon, however in most countries these are small.”

We have split Table 1 into Tables 1 (UN estimated births) and 2 (GBD estimated births), to make the data presentation clearer.

3) Around lines 113-118. Another important consideration is that in much of the global South, registration may also occur when (and be delayed until) a child needs to go to school, which may require proof of birth. However, such a delay obviously attenuates the utility of the registration data, since any child dying in the interim is not covered.

The author makes a good point here. We have now added a statement that an example of a delay in birth registration may occur because the birth is only registered when the child is about to commence school (Lines 113-114). We also state that the impact of child mortality on the completeness of birth registration data is more likely where registration is delayed (Lines 122-123).

Minor: Line 501. Months. Not years.

We have corrected this error.

Reviewer #3: The objective of this paper is to assess the reliability of self-reported estimates of birth registration completeness obtained from surveys. It compares self-reported estimates with estimates computed using birth registrations reported by a national authority and estimates of the number of live births (UN and GBD).

This paper is interesting and relevant. It is well written and clearly preented. Overall, I found the results plausible, but I am not (yet) entirely convinced that self-reported estimates overestimate birth registration completeness.

We thank the reviewer for their valuable comments on the manuscript.

I would encourage you to better discuss the impact of the denominator on estimates of completeness calculated from CRVS systems. In some countries, the numbers of live births are estimated using the number of births reported in CRVS. So, the denominator and the numerator are not independent. It will probably be found in places with high level of completeness, so it should not alter your results, but I think it is worth mentioning.

The reviewer makes a valid point. We have now added (in Lines 202-204) that “The GBD and UN do use birth registration as a source of fertility estimates where such data are complete, which may create dependence between the numerator and denominator; we therefore filter our analyses to countries with completeness less than 95% (see below).” 

“In countries with incomplete birth registration, both the UN World Population Prospects and GBD estimate live births predominantly from census and survey data using demographic and statistical models.”

 In other countries, numbers of live births are based mainly on fertility estimates and estimates of population size by age groups, which may not be very reliable. In case numbers of births are overestimated, this could lead to lower estimates of completeness. Again, it may not substantially influence your results, but you could discuss this possibility.

The reviewer makes a good point, similar to Reviewer #2. In Lines 416-423 we have now added the following sentences about the limitations of GBD and UN birth estimates: “There is some uncertainty regarding their accuracy because their two sets of estimates are derived from different analytical approaches that result in higher completeness where GBD estimated births are used when compared with UN estimated births (because GBD commonly estimates lower births). Additionally, their estimates of total births are derived from age-specific fertility rates applied to age-specific population data of women of reproductive age that is also subject to uncertainty. As a result, there can be significant differences in estimated completeness between the two sources, as in Lebanon, however in most countries these are small.”

I also think you should discuss the differences you find in CRVS registration completeness between UN and GBD (Table 2). In some countries, differences are huge (e.g. Lebanon), and often are non-negligible (e.g. Bolivia, India, Colombia, Mexico, Philippines). This indicates that the estimates of the number of births is far from perfect.

This issue relates to our response to the previous comment – we recognize and state that there is uncertainty in the UN and GBD birth estimates (and therefore birth registration completeness estimates), hence we have presented both sets of estimates. As noted, this can result in large differences (as in Lebanon) but smaller differences elsewhere. The new text in Lines 416-423 addresses this issue.

You could also provide analyses without the outliers. With GBD estimates, Paraguay and Lebanon are clear outliers, and Paraguay is also an outlier with UN data. Actually, if you remove these two countries, differences remain, but are smaller. Some other countries with big differences are also very small, and it may be worth mentioning it.

We have already identified three countries with at least 30 percentage points difference between self-reported completeness and birth registration completeness (both UN and GBD) – Paraguay, Solomon Islands and Rwanda (Lines 307-310, 357-358) – two of these countries have populations of at least 7 million. The calculation of the median difference in completeness overcomes the potential distortion of results due to these outliers, but differences still remain: in countries with completeness less than 95%, the median difference is 9-10 percentage points and the average difference is 14 percentage points.

I did not understand why countries in Table 3 are not presented in the same way as in Table 2.

The data for the eight countries in Table 3 (now Table 4) were unpublished data and were made available to the authors through established collaborations with colleagues in these countries. Given their unpublished nature, we decided to only report absolute differences with self-reported completeness. We added this statement in Lines 193-94.

I also think you could report results at several points in time for the countries. If we find strong variations over time in either source, this would suggest there are some issues with the date. Moreover, since monitoring progress in birth registration is mentioned as important topic in the introduction, your results would be all the more relevant.

The reviewer makes a good point. However, for many countries there is only one data point of self-reported completeness. As more data become available, this suggested approach could be the subject of a future study.

More literature would be useful to understand in more detail why survey data may lead to overestimating completeness. Research by Hertrich and Rollet in Mali (in French) worked on self-reported estimates in census data, and found that these were overestimated because of a wrong understanding of the rules by a few interviewers. This may be relevant to your paper.

We thank the reviewer for making us aware of this research. We have now included this as a possible reason for over-estimation of self-reported completeness as well as a reference to this research (Lines 100-102).

Other comments

Could you mention the countries where estimated births are greater than registerd births? (line 197)

The countries where registered births exceed either GBD or UN estimated births is commonly due to the stochastic variation of national registered birth numbers when compared to the smoothed trends used in estimating national births.

The countries where the registered births exceed GBD estimated births are: 

• Armenia

• Cuba

• Egypt

• Panama

• Suriname

• Tunisia

The countries where the registered births exceed UN estimated births are: 

• Argentina

• Armenia

• Barbados

• Cuba

• Egypt

• Mongolia

• Montenegro

• Suriname

• Thailand

• Tunisia

• Ukraine

---

## [Editor Report · Decision Letter 1]

11 May 2021

How reliable are self-reported estimates of birth registration completeness? Comparison with vital statistics systems

PONE-D-21-03554R1

Dear Dr. Adair,

We’re pleased to inform you that your manuscript has been judged scientifically suitable for publication and will be formally accepted for publication once it meets all outstanding technical requirements.

Kind regards,

Bernardo Lanza Queiroz, Ph.D

Academic Editor

PLOS ONE
---

## [Editor Report · Acceptance letter]

31 May 2021

PONE-D-21-03554R1 

How reliable are self-reported estimates of birth registration completeness? Comparison with vital statistics systems 

Dear Dr. Adair:

I'm pleased to inform you that your manuscript has been deemed suitable for publication in PLOS ONE. Congratulations! Your manuscript is now with our production department. 

Kind regards, 

on behalf of

Dr. Bernardo Lanza Queiroz 

Academic Editor

PLOS ONE